



# Wind tunnel comparison of four VAWT configurations to test load-limiting concept and CFD validation

Jan Wiśniewski[1], Krzysztof Rogowski[1], Konrad Gumowski[1], Jacek Szumbarski[1]

[1]Institute of Aeronautics and Applied Mechanics, Warsaw University of Technology, Warszawa, 00-665, Poland

*Correspondence to*: Jan Wiśniewski (*jhwisniewski@meil.pw.edu.pl*)

**Abstract.** The article describes results of experimental wind tunnel testing of four different straight bladed vertical axis wind turbine model configurations. The experiment tested a novel concept of vertically dividing and azimuthally shifting a turbine rotor into two parts with a specific uneven height division in order to limit cycle amplitudes and average cycle values of bending moments at the bottom of the turbine shaft to increase product lifetime, especially for industrial scale turbines.

Testing reduction effects of simultaneously including a vertical gap between turbine rotor levels, increasing shaft length but also reducing aerodynamic interaction between rotor levels, has also been performed. Experiment results have shown very significant decreases of bending moment cycle amplitudes and average cycle values, for a wide range of measured wind speeds, for dual-level turbine configurations as compared to a single-level turbine configuration. The vertical spacing between levels equal to a blade's single chord length has proven to be sufficient, in laboratory-scale, to limit interaction

between turbine levels in order to achieve optimal reductions of tested parameters through an operating cycle shift between two position-locked rotor levels during a turbine's expected lifetime. CFD validation of maintaining the effect in industrial scale has been conducted, confirming the initial conclusions.

## 1. Introduction

Vertical axis wind turbine (VAWT) blades, unlike horizontal axis wind turbine blades, work in a high range of angles of

attack within each rotation cycle (Ahmadi-Balouaki et al., 2014) . Resulting high amplitudes of bending moment values and high maximum moment values at the bottom of a wind turbine shaft in a rotation cycle (Galinos et al., 2016)  are strong deterrents to development of economically feasible large-scale VAWTs. The proposed concept for limiting those factors focuses on separating the rotor vertically into two or more parts of different lengths, shifted azimuthally in such a way as to maximally reduce maximum moment values and amplitudes at the bottom of the rotor shaft in a rotation cycle. The tested

case had two rotor-levels – a longer one, closer to the bottom of the shaft, and a shorter one, further from the bottom of the shaft; in order to achieve comparable values of bending moments at the bottom of the rotor shaft from each rotor-level. Additional spacing between rotor-levels was also tested in order to limit interaction between separate rotor-levels.

The topic of lift-based large-scale VAWTs, despite the aforementioned technological drawbacks – a solution to which can be seen tested below, has been met with resurfacing interest, due to their specific advantages. While factors related to the blade





tip not moving faster relative to the rest of the blade, allowing for lower noise emissions (Iida et al., 2004), lower bird death rates and no ice block launching, in areas where the risk exists, as compared to horizontal axis wind turbines (HAWTs) are important advantages for certain siting conditions, the key factor that keeps drawing researchers to VAWTs is the high aerodynamic efficiency potential. A study by Simão Ferreira et al. (Simão Ferreira et al., 2014), comparing 6 different methods for assessing power coefficients (Cp) for a wide range of tip speed ratio and rotor solidity, has calculated large-scale VAWT Cp for advantageous configurations for each model to be between 0.54 and 0.6. Straight bladed VAWTs specifically hold an additional advantage; that is easier manufacturing blades (Chinchilla et al. 2011).

## 2. Test case description

The testing was conducted in the WUT Variable Turbulence Tunnel in the 2.5 m wide and 2 m tall environmental test section of the tunnel.

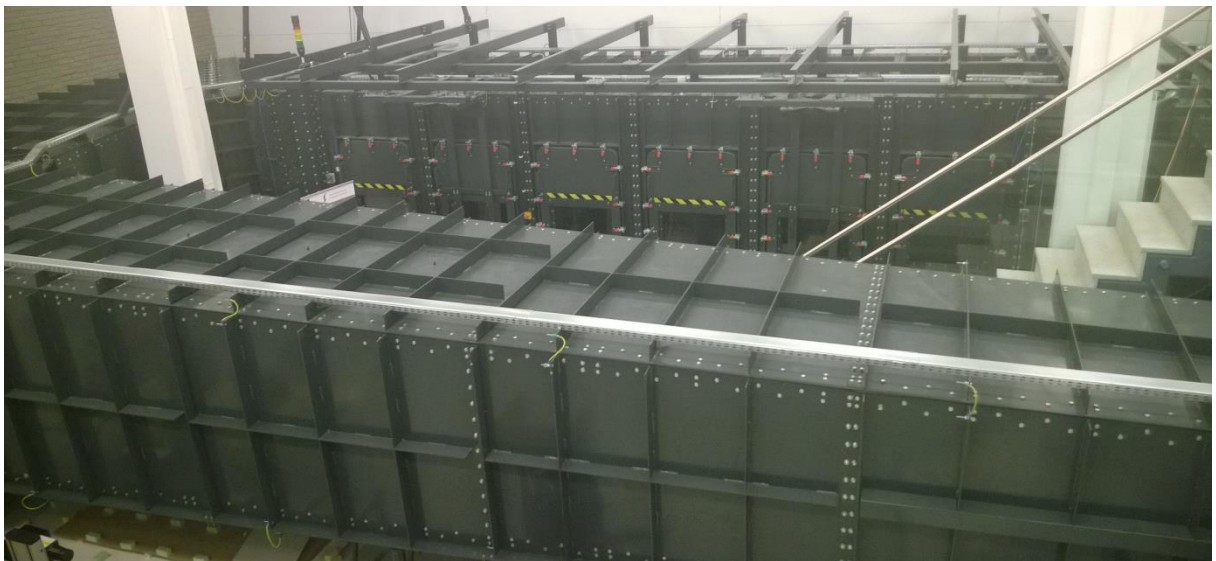

**Figure 1. WUT Variable Turbulence Tunnel**

Shown in Figure 1. the WUT Variable Turbulence Tunnel is a large-scale sub-sonic wind tunnel with two independent test sections allowing for testing in a range of speeds between 5 and 90 m/s. Air flow is generated by a 250 kW engine. Figure 1. shows a part of the Warsaw University of Technology including a section of the environmental part of the tunnel.

The model itself was 1.5 m high in the 2 shortest configurations, with a 57.5 cm turbine rotor diameter. The model was created with the upper level capable of shifting, in order to enable testing of different configurations. A three blade rotor





design was chosen, in addition to CFD simulations by the authors a comparative analysis by Parashivoiu shows them to have better structural reliability than dual blade designs (Parashivoiu, 2002). The model used the NACA 0018 symmetrical airfoil,

a classical VAWT airfoil used both in CFD based studies (Rogowski et al., 2018) and experiments (Laneville and Vittecoq, 1986).

The concept tested has, in the presented version, two levels of blades within a rotor. The lower level is equipped with longer blades in order to provide similar maximum and minimum values of bending moment at the bottom of the turbine shaft as the higher level, only shifted in cycle by azimuthally displacing the upper and lower rotor-level. In order to limit interference

influencing the character of aerodynamic loading on each three blade rotor level, variants with vertical spacing between rotor levels have also been tested – negatively influencing shaft length, but decreasing aerodynamic interference between adjacent rotor-levels. It is worthwhile to note that applying construction solutions reducing blade chord near the end of a rotor level (Islam et al., 2008), should result in lower spacing needs between adjacent turbine levels.

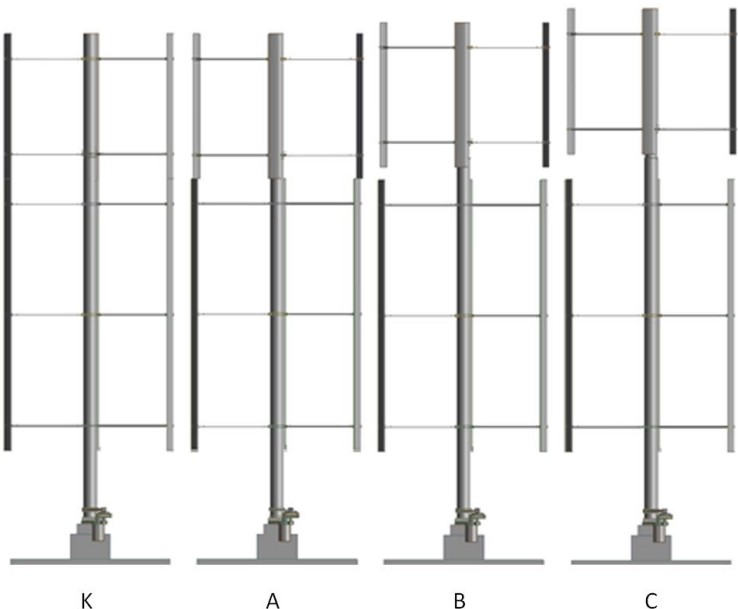

**Figure 2. Chosen VAWT configurations tested in the wind tunnel.**

Figure 2 displays the four configurations used for final wind tunnel testing. Configuration K is a standard single level VAWT with blade length equal to the sum of the length of both levels in other scenarios. Configuration A is a dual level wind turbine, shifted azimuthally by 60 degrees, with the second level starting at the exact height at which the first level



ends. Configuration B is analogous to A, whereas there is a vertical gap between adjacent levels equal to a single chord length (3.75cm). Configuration C has a vertical gap between levels equal to two chord lengths (7.5cm).

The reason for conducting the tests was to measure the bending moment values at the bottom of the turbine tower for a laboratory-scale model of the authors' analyzed turbine concept, within a few configurations. The values were being measured for a range of inflow wind speeds between 4m/s and 12m/s while the turbine was rotating freely. Each

measurement consisted of 10,000 data acquisitions within the period of 10 seconds. For many conditions up to 6 measurements were taken to ensure that a momentarily effect didn't influence the results. To reduce noise within the measured signal techniques from exploratory data analysis (EDA) were used (Oerlemans and Migliore, 2004). Firstly a technique called hanning, or a running weighted mean, was implemented. Each data point was replaced with the sum of half the data point and one-fourth of the previous and next data points. This was used consecutively three times for better results,

with 50-point median smoothing used twice afterwards for a final smoothened data set.

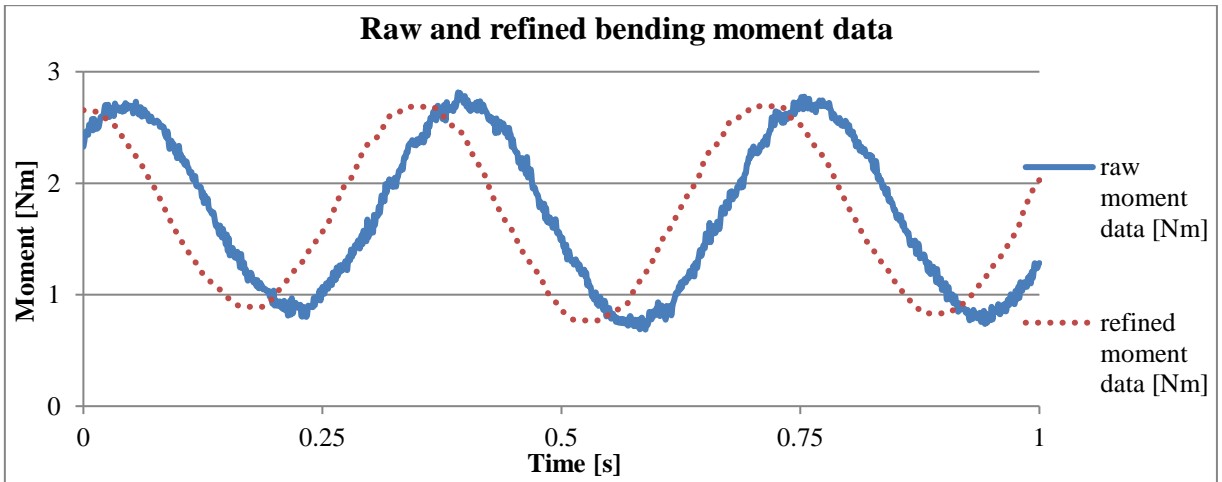

**Figure 3. Sample of raw and refined bending data values from configuration A**

Figure 3 presents a one second sample of raw bending moment values recorded by the tensometric scale used in the

experiment, as well as the refined bending moment values achieved by a five step smoothening process used in order to eliminate signal noise. The data is from a measurement in configuration A, taken at the inflow wind speed of 11 m/s. The necessity of eliminating signal noise does, to a small extent influence experiment results. The smoothening process, if done too subtly, maintains some artificial peak value increases. If the smoothening process is too major, it leads to chamfering of peak values resulting from actual physical forces acting upon the rotor. Although the smoothening was done with care, it is

important to remember that, especially for comparison of tens of cycles performed for four different geometries and a range



of wind speeds, it introduces risk of slightly altering peak values. While the process performed has no influence on the general nature of the experiment results or conclusions unto the effectiveness of the proposed solution, it is entirely possible it has a very slight influence on the exact result values.

**Configuration K**

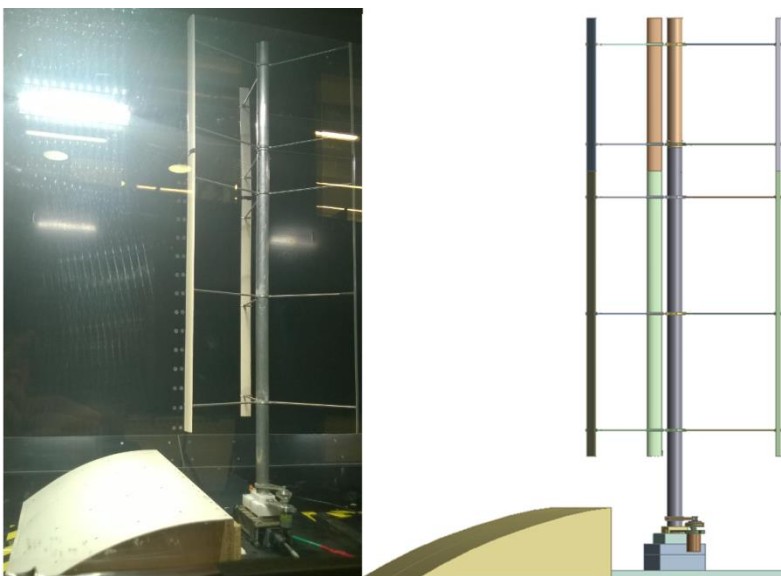


**Figure 4. Model set in configuration K**

Figure 4. displays the model set in configuration K. It shows a single level turbine, with straight, prolonged blades.

**Table 1.** Control case K testing parameters of bending moments for wind speed range

| Wind speed [m/s] | 4 | 5 | 6 | 7 | 8 | 9 | 10 | 11 | 12 |
|---|---|---|---|---|---|---|---|---|---|
| Average moment amplitude [Nm] | 0.077 | 0.102 | 0.243 | 0.358 | 0.522 | 1.193 | 7.799 | 3.316 | 1.672 |
| Max moment amplitude [Nm] | 0.118 | 0.183 | 0.389 | 0.604 | 0.809 | 1.529 | 8.249 | 3.762 | 2.326 |
| Min moment amplitude [Nm] | 0.025 | 0.021 | 0.036 | 0.168 | 0.157 | 0.833 | 7.211 | 2.701 | 1.066 |
| Average peak value [Nm] | 0.254 | 0.413 | 0.643 | 0.886 | 1.221 | 1.817 | 5.647 | 3.505 | 3.047 |
| Max peak value [Nm] | 0.270 | 0.509 | 0.753 | 0.994 | 1.363 | 1.976 | 5.941 | 3.727 | 3.414 |



Table 1 shows results of measuring bending moments at the bottom of the turbine tower for a range of wind speeds between 4 m/s and 12 m/s for a freely rotating wind turbine in configuration K. In general there is growth of bending moment values and amplitudes, accompanying the growth of inflow wind speeds. The growth, while not exactly proportional to the second power of the inflow speed, is strongly influenced by it, with two exceptions. It is noticeable from the values that at 10 m/s and  11 m/s, the turbine has started oscillating. It is especially evident for 10 m/s when the moment  amplitude is several
times larger than for many other measurements. It is also visible that for 10 m/s the moment  amplitude is noticeably higher than the moment  peak values, meaning that for a part of the loading cycle the turbine is being pushed forward against the direction of the wind. That effect is, to a much smaller degree visible at the inflow speed of 11 m/s.

## 2.2. Configuration A

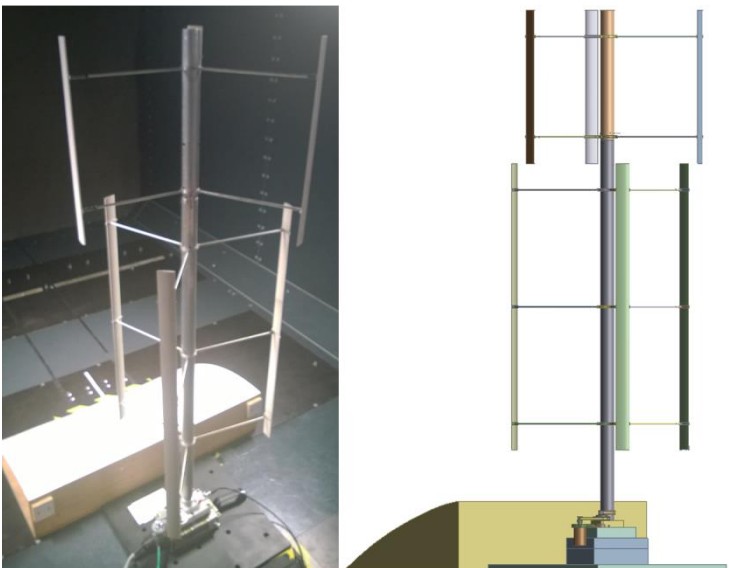

**Figure 5. Model set in configuration A**

Figure 5. displays the model set in configuration A. It shows a basic dual-level turbine, shifted azimuthally between levels by 60 degrees. There is no vertical displacement between levels – the upper level of the rotor starts at the same height the lower level ends.




**Table 2.** Case A testing parameters of bending moments for wind speed range

| Wind speed [m/s] | 4 | 5 | 6 | 7 | 8 | 9 | 10 | 11 | 12 |
|---|---|---|---|---|---|---|---|---|---|
| Average moment amplitude [Nm] | 0.058 | 0.133 | 0.133 | 0.170 | 0.179 | 0.258 | 0.521 | 1.917 | 0.721 |
| Max moment amplitude [Nm] | 0.098 | 0.199 | 0.235 | 0.262 | 0.336 | 0.508 | 0.898 | 2.132 | 1.390 |
| Min moment amplitude [Nm] | 0.025 | 0.084 | 0.052 | 0.084 | 0.043 | 0.108 | 0.214 | 1.710 | 0.188 |
| Average peak value [Nm] | 0.248 | 0.401 | 0.564 | 0.762 | 0.978 | 1.278 | 1.708 | 2.730 | 2.455 |
| Max peak value [Nm] | 0.278 | 0.441 | 0.612 | 0.833 | 1.145 | 1.394 | 1.922 | 2.893 | 2.854 |

Table 2 shows results of measuring bending moments at the bottom of the turbine tower for a range of wind speeds between
4 m/s and 12 m/s for a freely rotating wind turbine in configuration A. In general there is growth of bending moment values
and amplitudes, accompanying the growth of inflow wind speeds. At the inflow speed of 11 m/s, the shape of the curve as
well as the fact that both moment amplitudes and peak values are greater than for 12 m/s, suggest that for 11 m/s resonance
occurs. The growth of values due to resonance is much smaller than in the single-level configuration K.

**2.3 Configuration B**

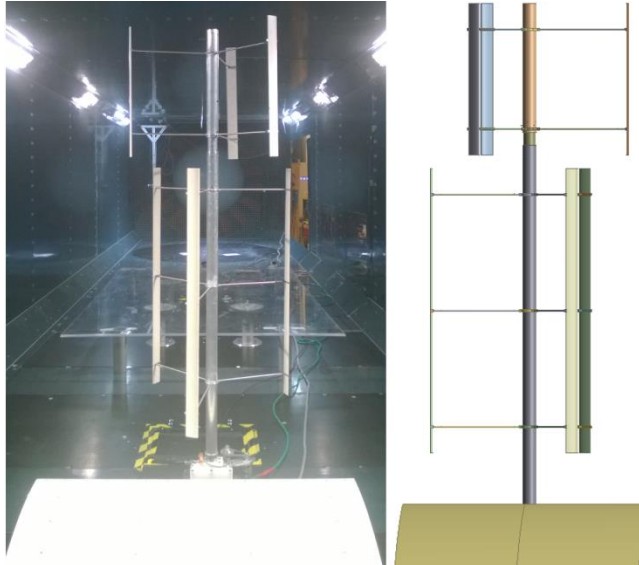

**Figure 6. Model set in configuration B**

Figure 6. displays the model set in configuration B. It shows a dual-level turbine, shifted azimuthally between levels by 60
degrees. The height of the vertical gap between rotor levels is equal to 3.75cm – 1 chord length.





**Table 3.** Case B testing parameters of bending moments for wind speed range

| Wind speed [m/s] | 4 | 5 | 6 | 7 | 8 | 9 | 10 | 11 | 12 |
|---|---|---|---|---|---|---|---|---|---|
| Average moment amplitude [Nm] | 0.059 | 0.065 | 0.097 | 0.145 | 0.177 | 0.411 | 0.399 | 1.223 | 0.595 |
| Max moment amplitude [Nm] | 0.088 | 0.113 | 0.157 | 0.259 | 0.274 | 0.645 | 0.664 | 1.480 | 0.891 |
| Min moment amplitude [Nm] | 0.025 | 0.022 | 0.030 | 0.043 | 0.066 | 0.185 | 0.164 | 0.895 | 0.128 |
| Average peak value [Nm] | 0.246 | 0.365 | 0.544 | 0.775 | 1.002 | 1.399 | 1.659 | 2.401 | 2.448 |
| Max peak value [Nm] | 0.262 | 0.397 | 0.575 | 0.844 | 1.063 | 1.562 | 1.820 | 2.498 | 2.583 |

Table 3 shows results of measuring bending moments at the bottom of the turbine tower for a range of wind speeds between 4 m/s and 12 m/s for a freely rotating wind turbine in configuration B. At the inflow speed of 11 m/s, the shape of the curve as well as the fact that both moment amplitudes and peak values are greater than for 12 m/s, suggest that for 11 m/s resonance occurs. The growth of values due to resonance is much smaller than in the single-level configuration K, and also lower than for configuration A, which has no vertical spacing between levels.

## 2.4. Configuration C

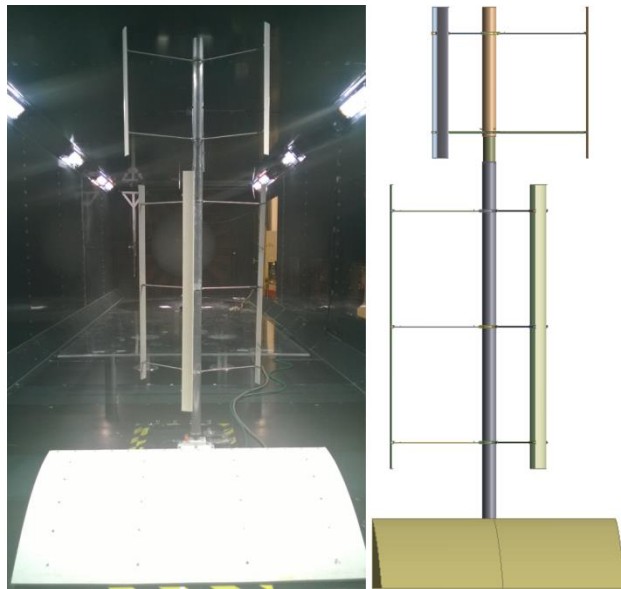


**Figure 7. Model set in configuration C**



Figure 7. displays the model set in configuration C. It shows a dual-level turbine, shifted azimuthally between levels by 60 degrees. The height of the vertical gap between rotor levels is equal to 7.5cm – 2 chord lengths.

**Table 4.** Case C testing parameters of bending moments for wind speed range

| Wind speed [m/s] | 4 | 5 | 6 | 7 | 8 | 9 | 10 | 11 | 12 |
|---|---|---|---|---|---|---|---|---|---|
| Average moment amplitude [Nm] | 0.063 | 0.090 | 0.147 | 0.136 | 0.249 | 0.290 | 0.393 | 0.940 | 0.580 |
| Max moment amplitude [Nm] | 0.111 | 0.149 | 0.209 | 0.217 | 0.441 | 0.520 | 0.766 | 1.264 | 1.122 |
| Min moment amplitude [Nm] | 0.029 | 0.055 | 0.072 | 0.059 | 0.077 | 0.035 | 0.100 | 0.630 | 0.144 |
| Average peak value [Nm] | 0.261 | 0.407 | 0.594 | 0.797 | 1.087 | 1.399 | 1.723 | 2.340 | 2.529 |
| Max peak value [Nm] | 0.287 | 0.448 | 0.643 | 0.864 | 1.159 | 1.564 | 1.946 | 2.490 | 2.776 |


Table 4 shows results of measuring bending moments at the bottom of the turbine tower for a range of wind speeds between 4 m/s and 12 m/s for a freely rotating wind turbine in configuration C. At the inflow speed of 11 m/s, the shape of the curve as well as the fact that both moment amplitudes and peak values are greater than for 12 m/s, suggest that for 11 m/s resonance occurs. The growth of values due to resonance is much smaller than in the single-level configuration K, and also

lower than for configuration A or B. The growth of vertical spacing between rotor levels, tested to quantify the effects of decreasing interaction between levels on bending moment values in addition to the azimuthal shift of rotor levels, helps limit the effects of the sudden moment and moment amplitude growth at certain wind speeds.

## 3. Experiment result Comparison







**Table 5.** Result comparison for different scenarios

| Inflow velocity [m/s] | Average bending moment amplitude reduction | | | | Average peak moment value reduction | | | |
|---|---|---|---|---|---|---|---|---|
| | K | A | B | C | K | A | B | C |
| 4 | 0% | 25% | 23% | 18% | 0% | 3% | 3% | -2% |
| 5 | 0% | -31% | 37% | 12% | 0% | 3% | 12% | 1% |
| 6 | 0% | 45% | 60% | 40% | 0% | 12% | 15% | 8% |
| 7 | 0% | 53% | 59% | 62% | 0% | 14% | 12% | 10% |
| 8 | 0% | 66% | 66% | 52% | 0% | 20% | 18% | 11% |
| 9 | 0% | 78% | 66% | 76% | 0% | 30% | 23% | 23% |
| 10 | 0% | 93% | 95% | 95% | 0% | 70% | 71% | 69% |
| 11 | 0% | 42% | 63% | 72% | 0% | 22% | 32% | 33% |
| 12 | 0% | 57% | 64% | 65% | 0% | 19% | 20% | 17% |

Table 5 shows the reduction of average bending moment amplitude and average peak bending moment value results for each separate inflow speed for every test configuration as compared to configuration K. Except for 4 m/s, all average bending

moment amplitude reduction levels for configuration B are greater than for configuration A. Configuration C shows superior average bending moment amplitude reduction levels to configuration B for 7 m/s, and the range of 9-12 m/s. The vertical spacing between levels also increases the height of the structure as well as moves the model further from optimal level-length proportions, which were optimized for configuration A. The results show that growth of vertical spacing corresponded to a drop in average peak bending moment value reduction for inflow speeds of below 10m/s, at which point resonance

begins to influence test results.

For results presented in this paper, there can be several ways of assessing the average reduction in bending moment and bending moment amplitude values. The simplest way would be to take reduction percentage values from table 5 and make a simple average of them. For purposes relating to product lifetime, a more realistic approach would be to take an average, but discard the values at low inflow speeds – too small to influence turbine lifetime, as compared to values at higher inflow

speeds. For a range of relevant wind speeds set from 8m/s to 12m/s, an average reduction of bending moment amplitude in configuration A was at 67%, while the average reduction of peak bending moment values was at 32%. For configuration B, the reductions calculated thusly were likewise 71% and 33%, and for configuration C – 72% and 31%. Another simple



approach would be to make an unweighted sum of all measured mean values for every configuration and quantify the reduction between those sums. A modification of this approach is weighing the results at all tested wind speeds, by

probability of their occurrence. This has been done using the Weibull wind speed distribution curve for reasonable European wind farm siting conditions – a middle-of-rotor average wind speed of 5.7 m/s and k shape factor equal to 2.1 (Kiss P, Jánosi I M, 2008). For this averaging method, moment amplitudes in configuration A were limited on average by 80%, while the average reduction of peak bending moment values at the tower base was 42%, as compared to configuration K. For configuration B, the reductions were likewise 82% and 42%, and for configuration C – 84% and 40%.

## 4. CFD validation


A validation of the load-limiting concept in industrial scale has been performed using 3D CFD in ANSYS Fluent. Compared was a dual-level straight-bladed wind turbine with the vertical spacing between levels equal to 1 blade chord length, based on experimental configuration B – 1.5m and a single level straight-bladed turbine with identical chord, rotor length and diameter.

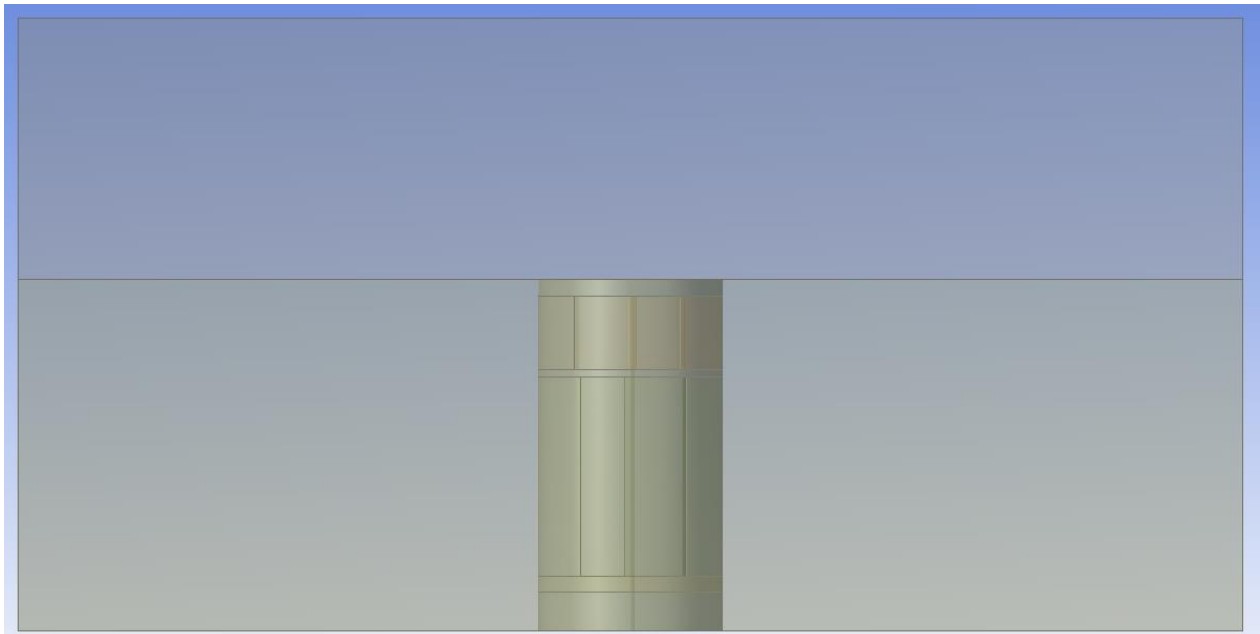


**Figure 8. Computational domain geometry for dual-level scenario**

Figure 8 shows a side view of the geometry used for the computational domain for the dual-level scenario. For both scenarios the blade chord, total rotor length and diameter are exactly 40 times that of the experimental cases. The blades were set 3 degrees to the outside of the rotor, relative to the blades' motion path. This parameter, and others such as the





chord to diameter ratio were chosen as a result of 2D CFD production optimization. The airfoil used, after testing the effect of airfoil thickness for a range of angles of blade attachment with 2D simulations, was once again NACA0018.

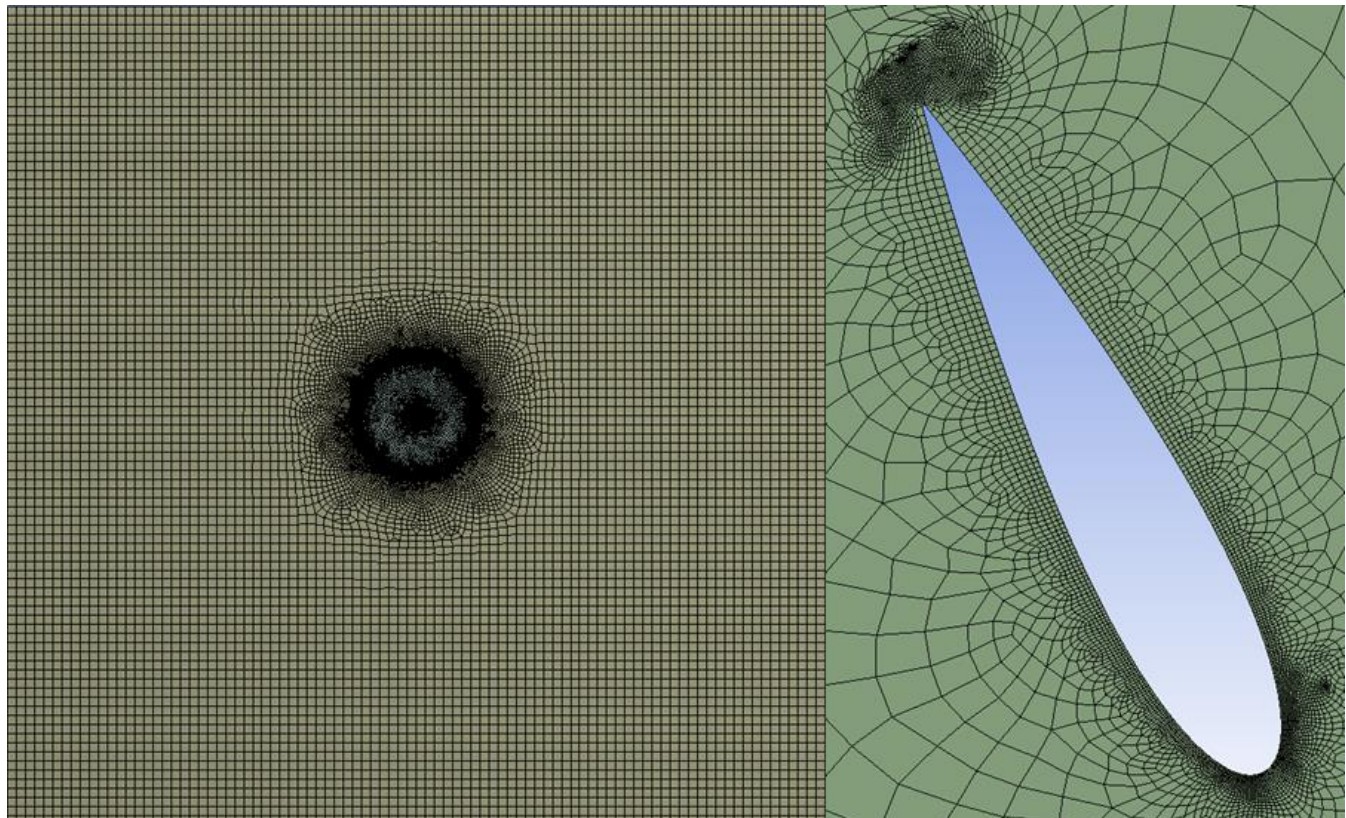

**Figure 9. Bottom side view of select mesh parts**

Figure 9 displays a bottom side view of the mesh for the entire domain as well as the sweepable mesh on one of the blades.
The trailing edge was divided into two parts, the rest of the blade into 550 parts. Automatic boundary layer creation – as incompatible with the sweep method used, was not implemented. The simulation was conducted with the k-omega SST turbulence method, default turbulence parameters and surface roughness and a 9 m/s inflow speed, a rotational speed of 140 deg/s and 0.01s time step.

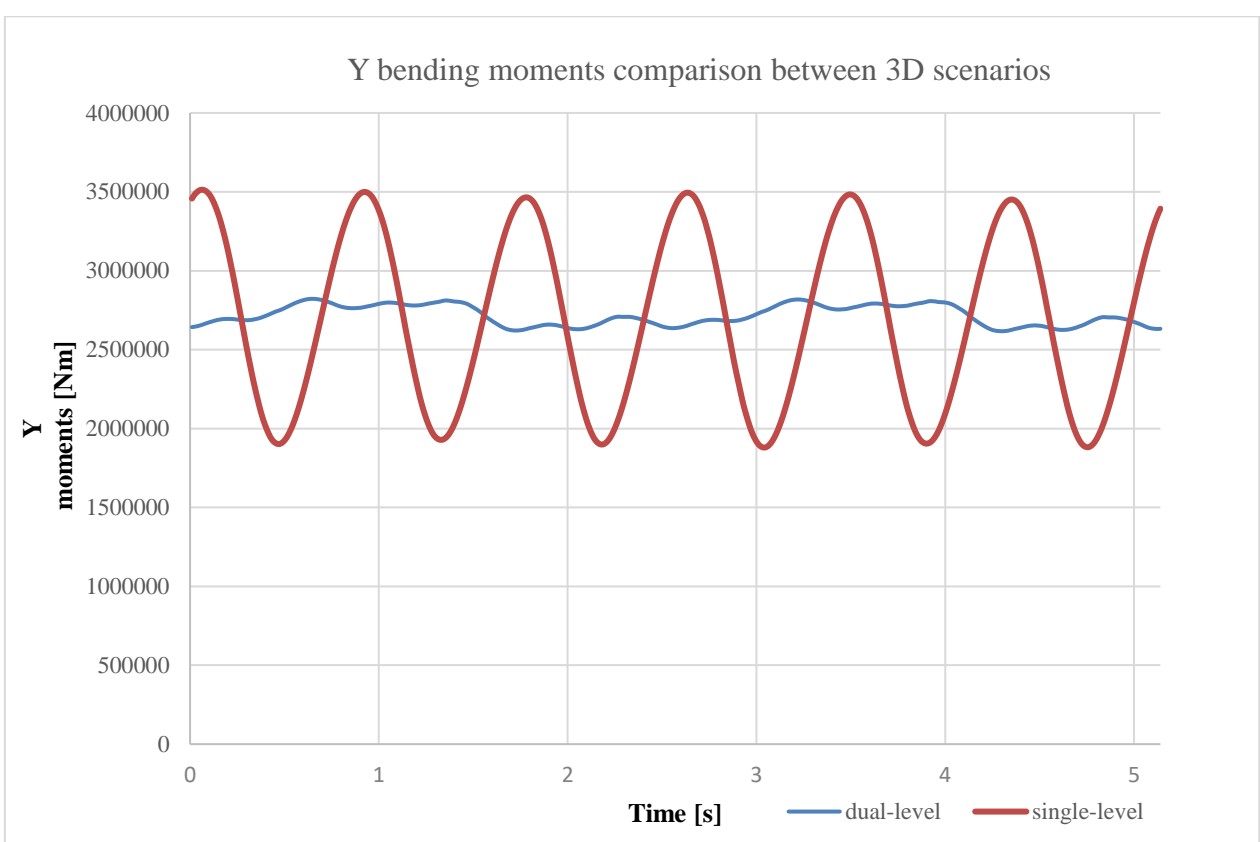

Figure 10. Y-bending moments comparison between 3D scenarios

Figure 10 shows the Y-bending moments, according to Fluent's default coordinate system, at the bottom of the rotor shaft, analogous to the moments measured during the experimental comparison in the first part of the paper. Compared to the single-level scenario, in the dual-level scenario the maximum Y-moment values within a cycle are limited by 19.7%, while the Y-moment amplitude is limited by 87.5%.



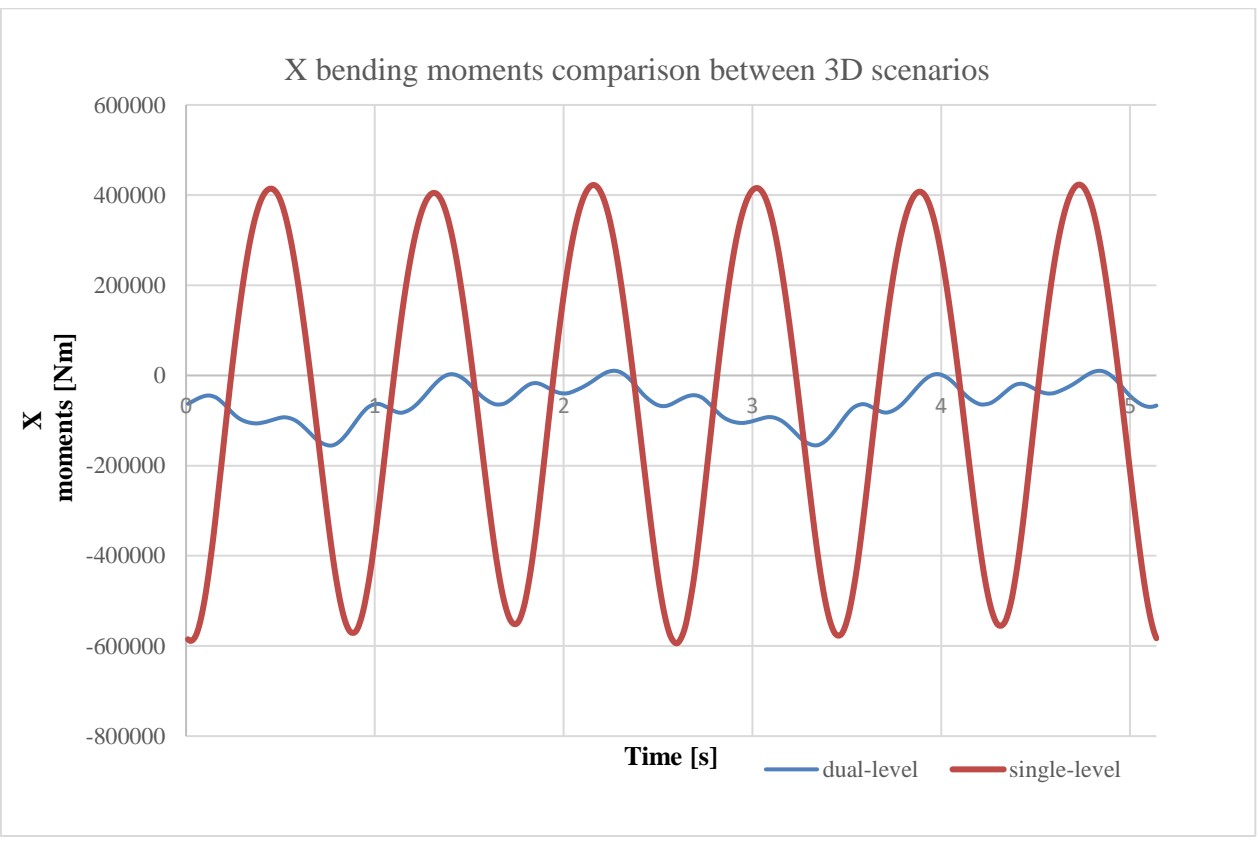

**Figure 11. X-bending moments comparison between 3D scenarios**

Figure 11 shows the X-bending moments at the bottom of the rotor shaft, generated due to lift. Compared to the single-level scenario, in the dual-level scenario the maximum positive X-moment values within a cycle are limited by 97.5%, the maximum negative X-moment values within a cycle are limited by 73.6%, while the X-moment amplitude is limited by 83.6%. Finally, in the dual-level scenario, the maximum total moments at the bottom of the rotor shaft are limited by 20.6%, while the total moment amplitude is limited by 87.4%.

## 5. Conclusions

In the laboratory scale model the lift-based bending moment component at the bottom of the turbine shaft became lost and unmeasurable among the measurement noise and a very unfavorable lift to drag ratio of the NACA0018 airfoil at low Reynolds numbers. Within the simulations, the 2.4 million to 4 million Reynolds numbers were much more advantageous in terms of airfoil lift to drag ratios resulting in a lift component taking a more distinct role in total bending moment values at





the bottom of the rotor shaft. The influence of the lift based component on total moments is much lower than the Y-component even in the large scale simulation, resulting in a small increase of maximum total moment value limiting, and a

slight decrease in limiting the total amplitude within a cycle. Both the experimental testing and large-scale CFD validation offered very high levels of reduction of bending moments at the bottom of the turbine shaft, proportional to cyclic loading values. The obtained results, along with prior tests, yield a high probability of the concept being applicable in creating reliable, sleeker and more cost-efficient designs than previously exploited. Further validating those assumptions with mid-scale environmental testing and a mechanical analysis of all relevant turbine elements is planned as the next research step

within the topic.

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
