# Peer review of "Wind tunnel comparison of four VAWT configurations to test loadlimiting concept and CFD validation"

_Wind Energy Science, 2020_

## Referee Comment (RC1) · Anonymous Referee #1 · 4 Jun 2020

The paper has a clear abstract with limited objectives enabling systematic investigation concerning VAWT rotor configurations and their influence on the cyclic bending moments seen at the base of the support tower/main shaft. In general nice connections are made in references to relevant previous work.

The swept area of a VAWT is in general rectangular. The configurations shown appear to have a height, H to diameter, D ratio of about 2.5. If the design tip speed and rated power are fixed and we compare with say height to diameter ratio, H/D of 1, the design with H/D= 2.5 has the advantage of lower rated torque but disadvantages of more blade area required and higher base moments. I don't think this impacts too much on your

study as I would expect that the results in terms of comparing loads in the K,A,B,C configurations would be similar at other H/D ratios. However this is not proven and it would be good to recognize it as another variable affecting in principle the generality of your conclusions.

There is no mention of spiral bladed VAWTs. The idea that distributing the position of blade elements around the rotor circle will smooth torque and loads is already well appreciated and this should be acknowledged. The spiral bladed VAWT is the ultimate in that respect doing it continuously. Your study is a special case where the distribution is in only two discrete blade sets. The case for your idea could then be that while the spiral blades are structurally efficient at small scale, they would be problematic at large scale.

As a general point on presentation, ahead of section 4, I find too many figures showing the configurations. In the space to the right of Fig 2 for example two vertical schematics of 3 blades for K and 6 for A,B,C would show all the configurations more clearly. Perhaps 4,5,6 could be collapsed into 1 or 2 composite figures. On the other hand, figures with graphical display of the results of Tables 1,2, 3 and 4 would be rather helpful.

The model testing lacks mention of Reynolds number effects until line 209 starting the conclusions. The comment is out of place there . Its not really a conclusion and should be discussed with the experimental results. How low was Re or the range of Re in the model tests?

In line 99 "started oscillating" . What kind of oscillations, bending, torsion?

English in the paper is generally good but from line 114, the word "growth" is not at all wrong but reads rather strangely. Better is "increase in bending moment values". The way it is written "growth" sounds as if the increase is unusual behaviour when, until stall and unsteady effects occur most significantly, we would expect increase in moments (perhaps as square of wind speed). The graphical presentation of Table 2 would definitely help here. The mention of the effects of resonance here is not telling

us much with no definition of its nature or suggested explanations.

Finally in your conclusions I think it is pushing it to say more "cost efficient". The results show how bending moments can be reduced and this is certainly useful information for a designer that may assist design optimisation. In a fully engineered system it is unclear how the cross arm structures (sizes and drag impacts) for K will compare with the cross arm structures required for blades in a sense cut into two , A, B,C.
* * *

---

## Referee Comment (RC2) · Anonymous Referee #2 · 10 Jun 2020

Summary:

The manuscript discusses a set of experiments and CFD simulations examining various VAWT configurations to reduce the cyclic shaft loads produced by the standard Darreius-style, 3-bladed VAWT. The authors have spent a good deal of time generating the experimental and numerical results which may have applications to VAWT design and optimization.

General Comments:

The manuscript could benefit from several revisions. The first of these is to expand the introduction and review sections, as they stand the literature review is weak and

incomplete. Many studies have been conducted on the twisted-blade VAWT and these should be included in the review and motivation given as to why the current geometry was chosen.

There are a large number of typos throughout the manuscript (use of "effect" vs "affect" and "smoothening" instead of the correct "smoothing").

Overall, the data presentation could be improved throughout the paper. Results placed in large tables are difficult to interpret and force the reader to sift through various tables to make comparisons. These data sets should be plotted in an organized fashion.

The conclusions section needs revision as well. The comments on the Reynolds number are out of place, with no other mention of the effect of Re anywhere else in the manuscript. Furthermore, the experimental results (on which the bulk of the paper focuses) are barely discussed, which of the 4 designs performed the best? How close did the simulation and experiment data match?

Line Comments:

The statement on line 33 stating the "high aerodynamic efficiency potential" needs to be further justified besides the Ferreira 2014 paper. Many articles have also shown the lower aerodynamic efficiency of the VAWT as compared with the HAWT, some reference should be made to these. There are also other benefits to the VAWT design not mentioned such as insensitivity to wind direction and the ability to mount the generator near to the ground.

Line 46: What is the blockage of the model in the tunnel? Were any corrections made to the experimental data to account for the effect of flow acceleration? It appears that the simulations did not reproduce the walls of the tunnel, so some correction should be used.

Line 60: Figure 2 is very difficult to interpret. Can dimensions be added to each figure and perhaps reduce the shading of the 3D CAD models so that they show up more

clearly? The figure caption should also have a brief but clear description of the 4 test cases to aid the reader.

Line 70: "For many conditions up to 6" Use specific language, what does "many conditions" mean? Also in this same sentence "momentarily" should be "momentary" and "effect" should be "affect" (there are other instances of this in the rest of the manuscript).

Line 83: The use of "smoothening" is incorrect it should be "smoothing". This should be fixed throughout the paper.

Line 83: What is meant by the term "chamfering"? Again, please use technical and precise language in the discussion.

Line 86: The entire sentence "While the process performed has no influence on the general nature of the experiment results or conclusions unto the effectiveness of the proposed solution, it is entirely possible it has a very slight influence on the exact result values." Is self-contradicting. How can a process have no influence on the results but have an influence on the exact result values? Did you mean that it does not change the data trends? Please clarify and re-word.

Line 90: Table 1 should be made into a plot, there is no need to have tabulated data for these comparison points in the paper (similar comment for other data tables).

Line 96: Plot the data sets non-dimensionally with the tip speed ratio, what you will find is that forces/moments scale with the velocity squared so this result is not surprising.

Line 115: Shape of what curve?

Line 165: Some comments about how these results might scale up from the laboratory experiments to full-scale Reynolds numbers would be useful. Comments on the CFD Section: Why are these results (and the plotted data) not compared directly with the experimental results of the previous section? I recommend making new plots showing the comparison directly. The section title is "CFD Validation", but you have not validated anything because there is no comparison to the experiments.

Line 209: The conclusions section needs to be revised due to several issues. The first is the discussion of the Reynolds number which is not mentioned anywhere else in the manuscript (for instance, what is the Re of the experiment?) It is also not surprising that the performance of the 0018 was poor, it is an airfoil designed for high Reynolds numbers (3 million and above). Also, the conclusion section makes no mention of the 4 different configurations, which one was the best?

---

## Author Comment (AC1) · 24 Jul 2020

Thank you for the input. The reply comments on the points are being made in order.

"The paper has a clear abstract with limited objectives enabling systematic investigation concerning VAWT rotor configurations and their influence on the cyclic bendingmoments seen at the base of the support tower/main shaft. In general nice connections are made in references to relevant previous work."

Thank you very much.

"The swept area of a VAWT is in general rectangular. The configurations shown appear

to have a height, H to diameter, D ratio of about 2.5. If the design tip speed and rated power are fixed and we compare with say height to diameter ratio,H/D of 1, the design with H/D= 2.5 has the advantage of lower rated torque but disadvantages of more blade area required and higher base moments. I don't think this impacts too much on your study as I would expect that the results in terms of comparing loads in the K,A,B,C configurations would be similar at other H/D ratios. However this is not proven and it would be good to recognize it as another variable affecting in principle the generality of your conclusions."

Those are very good points. The 3D validated scenario is optimized in terms of cost-efficiency, however as the materials and cost analysis were performed as part of a non-published commercial outside study, this is a troubling matter reference-wise. You are very correct that this matter has to be addressed.

"There is no mention of spiral bladed VAWTs. The idea that distributing the position of blade elements around the rotor circle will smooth torque and loads is already well appreciated and this should be acknowledged. The spiral bladed VAWT is the ultimate in that respect doing it continuously. Your study is a special case where the distribution is in only two discrete blade sets. The case for your idea could then be that while the spiral blades are structurally efficient at small scale, they would be problematic at large scale."

Not mentioning spiral bladed VAWTs was an attempt at limiting the scope of the discussion and to avoid inadvertently leaking intellectual property too early. Right now I would love to add some content about spiral bladed VAWTs. However it must be noted that typical designs of such turbines are not optimal in smoothing bending moments – there is an effect, but as the upper sections have greater leverage than the lower ones the effect achieved is far from perfect. The solution to this, described in a soon to be published patent application PCT/PL2020/000054 lies in a non-linear twist – operating on a similar principle to the upper portion of the 2-part H-VAWT rotor being a specific different size than the lower portion. Finally, both structural concerns and increased weight

of spiral blades makes the technology very interesting in small to medium scales, but far less cost-efficient than the presented scenario in the scale presented and above.

As a general point on presentation, ahead of section 4, I find too many figures showing the configurations. In the space to the right of Fig 2 for example two vertical schematics of 3 blades for K and 6 for A,B,C would show all the configurations more clearly. Perhaps 4,5,6 could be collapsed into 1 or 2 composite figures. On the other hand, figures with graphical display of the results of Tables 1,2, 3 and 4 would be rather helpful. The model testing lacks mention of Reynolds number effects until line 209 starting the conclusions. The comment is out of place there. Its not really a conclusion and should be discussed with the experimental results."

Noted.

"How low was Re or the range of Re in themodel tests?"

Around 10 000 to 50 000 – a very poor range for symmetrical NACA characteristics.

In line 99 "started oscillating" . What kind of oscillations, bending, torsion?

Thank you for the comment – bending.

"English in the paper is generally good but from line 114, the word "growth" is not at all wrong but reads rather strangely. Better is "increase in bending moment values". The way it is written "growth" sounds as if the increase is unusual behaviour when, until stall and unsteady effects occur most significantly, we would expect increase in moments (perhaps as square of wind speed). The graphical presentation of Table 2would definitely help here. The mention of the effects of resonance here is not telling us much with no definition of its nature or suggested explanations."

Thank you for the corrections.

"Finally in your conclusions I think it is pushing it to say more "cost efficient". The results show how bending moments can be reduced and this is certainly useful information

for a designer that may assist design optimisation. In a fully engineered system it is unclear how the cross arm structures (sizes and drag impacts) for K will compare with the cross arm structures required for blades in a sense cut into two , A, B,C" While this hypothesis has proven to be true, it is based on unpublished outside work – I am very open and thankful for pointing out the issue and possible suggestions whether it is better to make the statement weaker as I do not think we have a right to reference the validation materials; or whether it is better to solve the issue some other way.

---

## Author Comment (AC2) · 24 Jul 2020

Thank you for the input. The reply comments on the points are being made in order "Summary: The manuscript discusses a set of experiments and CFD simulations examining various VAWT configurations to reduce the cyclic shaft loads produced by the standard Darreius-style, 3-bladed VAWT. The authors have spent a good deal of time generating the experimental and numerical results which may have applications to VAWT design and optimization."

Thank you.

[Figure]

"General Comments: The manuscript could benefit from several revisions. The first of these is to expand the introduction and review sections, as they stand the literature review is weak and incomplete. Many studies have been conducted on the twisted-blade VAWT and these should be included in the review and motivation given as to why the current geometry was chosen."

We would gladly expand on those points. If possible we would also be thankful for specific important articles that should not be missed within this part.

"There are a large number of typos throughout the manuscript (use of "effect" vs "affect" and "smoothening" instead of the correct "smoothing").Overall, the data presentation could be improved throughout the paper. Results placed in large tables are difficult to interpret and force the reader to sift through various tables to make comparisons. These data sets should be plotted in an organized fashion. The conclusions section needs revision as well. The comments on the Reynolds number are out of place, with no other mention of the effect of Re anywhere else in the manuscript."

Thank you for the corrections

"Furthermore, the experimental results (on which the bulk of the paper focuses) are barely discussed, which of the 4 designs performed the best?"

Truth be told which design performed the best changes depending on the criterion one might choose. For a general answer to be justified eg sets of cost-effectiveness studies of the tested designs would have to be generated. Design B was chosen as the favored one, being used as the basis for CFD validation – showing improvement in one of the desired reductions over A and no drop in the other as compared to A – like configuration C did. If that would be helpful we would explain that point in a revision of the article

"How close did the simulation and experiment data match?"

As the cases in the simulation and experiment were different, being made for entirely

different scales, we were concerned it might not be correct to compare them directly. It would be very interesting if one were to make a simulation in the same scale as the experiment and match the results. As the point was mainly to validate the design for large-scale

"Line Comments: The statement on line 33 stating the "high aerodynamic efficiency potential" needs to be further justified besides the Ferreira 2014 paper. Many articles have also shown the lower aerodynamic efficiency of the VAWT as compared with the HAWT, some reference should be made to these. "

The articles claiming lower aerodynamic efficiency are made in regard to small designs or low H/D ratio designs as high H/D ratios are problematic loading-wise, a problem that the tested hypothesis tries to solve. We can include the articles making statements based on less related cases, but are not sure whether it is correct to make a critical stance on the views expressed in them without a thorough shift in the article focus, to deeply explain the stance opposing some of the conflicting claims within different sources.

"There are also other benefits to the VAWT design not mentioned such as insensitivity to wind direction and the ability to mount the generator near to the ground."

Yes.

"Line 46: What is the blockage of the model in the tunnel?"

That is a very good concern – it is not nearly optimal for many purposes. Around 16% without the step before the rotor, 20% with.

"Were any corrections made to the experimental data to account for the effect of flow acceleration?"

No.

"It appears that the simulations did not reproduce the walls of the tunnel, so some

correction should be used."

It most probably should be more clearly stated within the article that the experiment and CFD case are not trying to show the exact same case, but closest available cases to experimentally and numerically validate the overall usefulness of the special large VAWT concept showcased within the article.

"Line 60: Figure 2 is very difficult to interpret. Can dimensions be added to each figure and perhaps reduce the shading of the 3D CAD models so that they show up more clearly? The figure caption should also have a brief but clear description of the 4 test-cases to aid the reader.Line 70: "For many conditions up to 6" Use specific language, what does "many conditions" mean? Also in this same sentence "momentarily" should be "momentary" and "effect" should be "affect" (there are other instances of this in the rest of the manuscript).Line 83: The use of "smoothening" is incorrect it should be "smoothing". This shouldbe fixed throughout the paper.Line 83: What is meant by the term "chamfering"? Again, please use technical and precise language in the discussion. "

Yes, thank you for the corrections.

"Line 86: The entire sentence "While the process performed has no influence on the general nature of the experiment results or conclusions unto the effectiveness of the proposed solution, it is entirely possible it has a very slight influence on the exact result values." Is self-contradicting. How can a process have no influence on the results but have an influence on the exact result values? Did you mean that it does not change the data trends? Please clarify and re-word."

Yes, thank you. Numerical and experimental values under specific conditions are not guaranteed to be the same in real life conditions, however they in no way invalidate the load-limiting hypothesis of the concept, rather showing very promising results. Further results based on outside non-published studies sadly cannot be referenced to further showcase this point.

"Line 90: Table 1 should be made into a plot, there is no need to have tabulated data for these comparison points in the paper (similar comment for other data tables)."

Future work based on results within other authors' articles cannot accurately be made based on plots

"Line 96: Plot the data sets non-dimensionally with the tip speed ratio, what you will find is that forces/moments scale with the velocity squared so this result is not surprising."

Yes

"Line 115: Shape of what curve?"

Please excuse us - bending moment data curve.

"Line 165: Some comments about how these results might scale up from the laboratory experiments to full-scale Reynolds numbers would be useful. Comments on the CFD Section: Why are these results (and the plotted data) not compared directly with the experimental results of the previous section? I recommend making new plots showing the comparison directly. "

That would be somewhat hard to explain as they are not related to the same case – rather they are two distortions of a large-scale real-life scenario that would be beneficial but extremely expensive to validate directly. Therefore the validation of possible advantages of the concept happens partially independently through two methods.

"The section title is "CFD Validation", but you have not validated anything because there is no comparison to the experiments."

Validation refers to the turbine concept. We will try to make that goal more clear within the article.

"209: The conclusions section needs to be revised due to several issues. The first is the discussion of the Reynolds number which is not mentioned anywhere else in the manuscript (for instance, what is the Re of the experiment?) It is also not surprising

that the performance of the 0018 was poor, it is an airfoil designed for high Reynolds numbers (3 million and above). Also, the conclusion section makes no mention of the 4 different configurations, which one was the best?"

Thank you for the thorough review, if it would be judged that with such corrections the article could be suitable for publishing we will very gladly clarify those points.